# Controllable Emphasis with zero data for text-to-speech

*Arnaud Joly, Marco Nicolis, Ekaterina Peterova, Alessandro Lombardi, Ammar Abbas,*
*Arent van Korlaar, Aman Hussain, Parul Sharma, Alexis Moinet, Mateusz Lajszczak,*
*Penny Karanasou, Antonio Bonafonte, Thomas Drugman, Elena Sokolova*

## Amazon, United Kingdom

{jarnaud,nicolism}@amazon.co.uk

## Abstract

We present a scalable method to produce high quality emphasis for text-to-speech (TTS) that does not require recordings or annotations. Many TTS models include a phoneme duration model. A simple but effective method to achieve emphasized speech consists in increasing the predicted duration of the emphasised word. We show that this is significantly better than spectrogram modification techniques improving naturalness by 7.3% and correct testers' identification of the emphasized word in a sentence by 40% on a reference female en-US voice. We show that this technique significantly closes the gap to methods that require explicit recordings. The method proved to be scalable and preferred in all four languages tested (English, Spanish, Italian, German), for different voices and multiple speaking styles.

**Index Terms**: text-to-speech, emphasis control

## 1. Introduction

Salient constituents in utterances, typically expressing new information, are intonationally focalized, bringing them to the informational fore. While the interaction between information structure and its acoustic correlates is nuanced and complex (see [1] for an overview), we follow [2] and much related work in characterizing narrow focus as affecting a single word/constituent, as opposed to broad/wide focus affecting the entire event denoted by the sentence. Consider the following examples from [2]:

(1)  a. Who fried an omelet?
     b. What did Damon do to an omelet?
     c. What did Damon fry?
     d. What happened last night?
     e. Damon fried an omelet.

(1e) is uttered with wide focus when it answers (1d), an out-of-the-blue context, and with a narrow focus when uttered as an answer to (1a-c): specifically subject focus in (1a), verb focus in (1b), object focus in (1c). The objective of this paper is to understand how we can provide "narrow focus" word-level emphasis controllability for multiple voices and languages (1) without quality degradation, (2) without annotation, (3) without recordings and (4) if possible without model re-training.

While context awareness of TTS system has vastly improved (see [3], [4] among others), automated output does not always assign the correct intonation to cases like (1e), given preceding context . Several commercial TTS system thus allow users to tweak the automated output by manually assigning emphasis (which we use as an umbrella term for narrow or contrastive focus) to a selected word.

A popular approach consists in recording a smaller dataset featuring the desired emphasis effect in addition to the main 'neutral' recordings, and having the model learn the particular prosody associated with the emphasized words (see [5, 6, 7, 8] for recent examples). We build one such model as our upper anchor, as detailed in section 2.1

While this technique works well for the speaker for which 'emphasis recordings' are available, it does not directly scale to new speakers or different languages. An alternative technique adopted with varying degrees of success consists in annotating existing expressive recordings for emphasis [9, 10, 11]; while this makes recordings not needed, scaling to new voices/languages is still expensive and time consuming, given the need for extensive annotation. Automatic annotation [12, 13, 14] could alleviate the issue, but these emphasis detectors rely on annotated data and there are no evaluation showing the generalization across datasets. In addition, given differing degrees of expressivity in different recordings, this approach is bound to work unevenly across different voices.

Recent developments in TTS research allow for explicit control of specific speech features (e.g. duration [15], [16], duration and pitch [17], etc.), thus providing the right tools to explicitly control acoustic features associated with emphasis in a voice-agnostic fashion, with no need for targeted recordings or annotations. Direct modification of speech features is of course an old idea in the field: for example, techniques based on TD-PSOLA [18] did allow for direct signal modification, but at a high cost in terms of quality / signal distortions [19]. A more modern incarnation of the idea is to directly modify the mel-spectrogram before vocoding. We adopt the latter approach as our baseline, as detailed in Section 2.4.

The very detailed study in [2] measured twelve acoustic features of focalized and non-focalized constituents and concluded that the top four dimensions characterizing focalization in English are as follows: (1) duration + silence (syllables duration longer for focalized words and silence longer before/after focalized word), (2) mean F0 (higher), (3) maximum F0 (higher), and (4) maximum intensity (higher). Other studies have largely confirmed the importance of these dimension cross-linguistically though ranking may differ (see e.g. [1] on German, where vowel lengthening ranked 8th out of 19 dimensions considered (unclear whether authors also considered silence associated with duration changes as in [2]). The issue of whether all four (or more) dimensions mentioned above are necessary to trigger the perception of emphasis has received somewhat marginal attention in the linguistics literature on the topic and is generally rather inconclusive (see e.g. [20] for the claim that an f0 rise is neither a necessary nor a sufficient condition for the perception of focus in Swedish).

The central claim we advance in this paper is that modelling

a duration increase of the phonemes belonging to the word targeted by emphasis (see below for details) suffices in most cases to trigger the perceptual impression of prominence. We show in section 3 that when the emphasis is perceptually particularly convincing, the model has implicitly learned to add silence before the syllable carrying main stress in the emphasized word, and f0 in the syllable carrying main stress shows a rising contour. We conclude that while this approach does not work perfectly in all cases, it may not be necessary to directly control all relevant acoustic dimensions to model emphasis, because models will tend to automatically correlate such dimensions, given context.

The cross-linguistic impact of this finding is broad: we expect most European languages to be amenable to the approach detailed in this paper. We report below positive results for English, German, Italian, Spanish.

The paper is organized as follows: section 2 introduces our TTS architecture, the baselines and it details our approach. In Section 3, we describe our evaluation methodology and empirical results both on English and other tested languages, providing cross-linguistic validity to our approach. Section 4 reports our conclusions and directions for future work.

## 2. Methods

### 2.1. Non-attentive TTS architecture

Our base TTS architecture (see Figure 1) is non-attentive with disjoint modelling of duration and acoustics. It is similar to DURIAN+ from [21], which is inspired by DURIAN [22] and FASTSPEECH [23]. The acoustic model aims to predict the mel-spectrogram sequence associated to a phoneme sequence. It consists of a TACOTRON2 [24] phomeme encoder, a phoneme-to-frame upsampler which is smoothed with a Bi(directional)-LSTM [25]. We train the acoustic model with oracle phoneme durations, also known as phoneme-to-frame alignment [26], extracted from the training data. In parallel, we train a duration model which will predict at inference time the duration of each phoneme given the phoneme sequence. The duration model as in [21, 27] consists of a stack of 3 convolution layers with 512 channels, kernel size of 5 and a dropout of $30\%$, a Bi-LSTM layer and a linear dense layer. To produce speech, we vocode the mel-spectrograms frame using a universal vocoder [28].

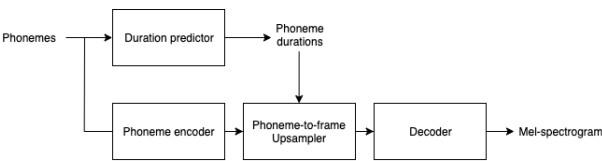

Figure 1: *Non-attention-based TTS architecture with external duration modelling.*

### 2.2. Datasets

All datasets mentioned in this paper are internal datasets recorded for the purpose of TTS voice creation. With the exception of the two-hour dataset mentioned in the next section in conjunction with the female-0 voice, no dataset was specifically recorded with the intention of obtaining emphatic speech. As pointed out below, different voices differ in terms of overall expressivity, as a results of the data used to train the model.

### 2.3. Emphasis through recordings

For our upper-bound system, we augmented our data with about 2hrs of additional recordings (of the female-0 voice), where the voice talent would read a sentence multiple times, each time with a different word emphasized. We modified the architecture of the TTS-system described in figure 1, by adding a word-level binary flag encoder (see Figure 2) both in the duration and acoustic models. The word level flag is upsampled to phoneme level: each phoneme in the utterance is thus effectively marked as either belonging to an emphasized word or not. It is then concatenated to the phoneme embedding that is the input to the phoneme encoder. This allows the model to create features combining both phoneme and emphasis level information. The model will imitate the provided recordings by modifying the prosody for the target word, and implicitly for the neighboring words. We will refer this approach as FLAG-EMPH.

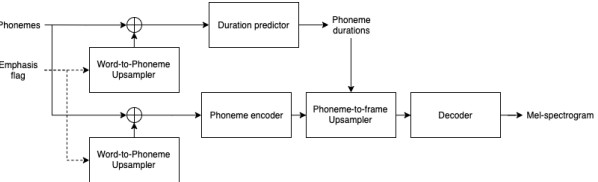

Figure 2: *TTS model with data driven word-level flag emphasis control with external duration modelling.*

### 2.4. Emphasis through speech mel-spectrogram modification

Most TTS generation systems are divided into two stages: (1) generation of Mel-spectrogram from a phoneme sequence, (2) creation of the waveform with a vocoder. Our baseline system MEL-EMPH produces word level emphasis by modifying the generated mel-spectrograms before vocoding by increasing the duration by a factor $\alpha_{mel} = 1.25$ and by increasing loudness amplitude by a factor $V_{mel} = 1.15$. These values were selected empirically as moderate emphasis level.

Increasing the loudness is obtainable by multiplicative scaling $V_{mel}$ of the mel-spectrogram frames. Duration control is achievable by modifying the upsampling factor from frame level ($80\,\mathrm{Hz}$) to waveform sample-level ($24\,\mathrm{kHz}$) [28, 29]. Each frame consists of $50\,\mathrm{ms}$ and are shifted by $12.5\,\mathrm{ms}$. For a speech at $24\,\mathrm{kHz}$, it corresponds to an upsampling by 300. Modifying this number by $\alpha_{mel}$ allows to control the duration.

### 2.5. Emphasis through model duration control

The proposed approach called Duration Dilatation emphasis (DD-EMPH) provides emphasis by modifying the duration of each phoneme before creating a mel-spectrogram. With non-attentive models, we have extracted the duration modelling from the mel-spectrogram generation. Our central claim is that it is possible to produce emphatic speech by lengthening the duration $d_p$ of each phoneme $p$ by a constant $\alpha_{DD}$ factor:

$$\hat{d}_p = \lceil \alpha_{DD} d_p \rceil, \tag{1}$$

where $\lceil \rceil$ is the ceiling operator, which make sure that lengthening happened when $\alpha_{DD} \in ]1.0, 1.5]$. In this paper, we will use $\alpha \in \{1.25, 1.5\}$. This approach can be applied to any non attentive TTS system, where the acoustic model is driven by the duration model.

By modifying the phoneme duration, we force the model to generate a modified sequence of mel-spectrograms. Our assumption is that it leads perceptually to emphasise the word. This approach is done only at inference time and does not require re-training.

We are aware that further improvements are achievable by carefully differentiating among different phoneme classes (see [30] for a linguistically grounded approach to duration modelling). However, current duration models appear to be able to correctly generalize, even in the absence of fine-grained subcategorizations. For example, while stops and affricates are obviously not very good candidates for lengthening, simply modifying durations in the acoustic model uniformly for all phonemes does not give rise to any artifacts, plausibly because the training data obviously does not contain any instance of 'long stop/affricate' to be learned.

# 3. Empirical analysis

## 3.1. Evaluation methodology

We evaluate two aspects of TTS with emphasis control: (1) the acoustic quality of the generated speech given the emphasis control, (2) the adequacy of the control policy, ie whether naïve native listeners can correctly identify which word was the one emphasized by our models.

We leveraged MUSHRA [31] whenever recordings are available and preference tests otherwise. For the MUSHRA test, we asked 24 listeners to "rate the naturalness between 0 and 100 of the presented voices considering that one word indicated in the text should sound emphasized". For preference test, we asked at least 50 listeners to "pick the voice they prefer considering one word as indicated in the text should sound emphasised". For these tests, we will show $\Delta Pref.$ the average fraction of listeners who voted for DD-EMPH against the other specified systems.

To assess identifiability, we asked 24 internal high performance internal professional listeners to identify which words are emphasised over 50 utterances, and computed the average fraction of time that the emphasized word was properly recognized as the most emphasised. Note that the listeners are not aware of which words are emphasised in this test.

We tested our approaches on a private internal dataset containing 7 voices in 4 locales in 3 styles with amount of recordings shown in Table 1. Voices were evaluated with native speakers with questions and utterances in the target language.

Table 1: *Available training data per voice.*

|  | Voice | Recordings [h] |
|---|---|---|
| en-US | female-0 exp. | 24 h highly expressive |
| en-US | male-0 conv. | 6 h conversational + 22h neutral |
| en-US | female-1 neutral | 31 h neutral |
| en-US | female-1 exp. | 12 h highly expressive |
| es-US | female-2 neutral | 29 h neutral |
| es-US | female-3 neutral | 28 h neutral |
| es-US | female-3 expressive | 24 h highly expressive |
| de-DE | female-4 neutral | 44 h neutral |
| it-IT | female-5 conv. | 6.6 h conversational + 26 h neutral |

## 3.2. Emphasis TTS system based on recordings

In this section, we would like to compare two models: the baseline MEL-EMPH and FLAG-EMPH, which is based on recordings. For this experiments, we recorded 1486 utterances for the reference voice "female-0 exp.". It correspond to a bit less of 2 hours of recordings with a single word that is emphasised. The voice talent is requested to bring narrow and focus emphasis on the emphasised word.

We observe in Figure 3 that having FLAG-EMPH improves over MEL-EMPH by 12.7% over the MEL-EMPH baseline. We tried to reduce the amount of data needed to produce high quality emphasis and observed in Figure 4 that it would need at least 1000 recorded utterances.

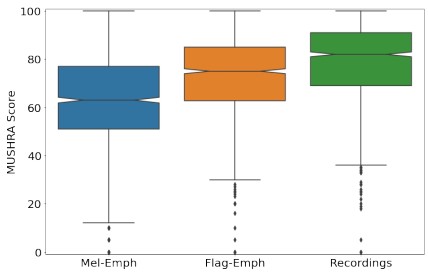

Figure 3: FLAG-EMPH *improves naturalness over* MEL-EMPH *with $p < 0.0001$ according to a Friedman test. Average MUSHRA score are 64.6 for* MEL-EMPH*, 72.8 for* FLAG-EMPH *and 78.1 for Recordings.*

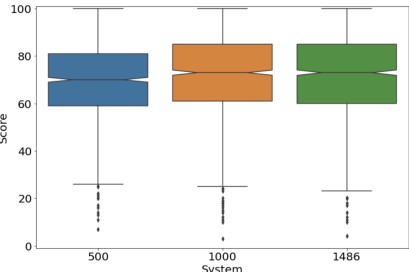

Figure 4: FLAG-EMPH *requires at least 1000 utterance to produce high quality emphasis for the voice female-0-exp. Average MUSHRA score of* FLAG-EMPH *are 69.1 for 500 utterances, 71.3 for 1000 utterances and 71.5 for 1486 utterances with $p < 0.0001$ according to a Friedman test.*

## 3.3. Dive deep on Emphasis TTS without recordings with DD-EMPH

For our reference voice "female-0 exp.", we observe on Figure 5 that DD-EMPH with $\alpha_{DD} = 1.5$ improves emphasis naturalness over MEL-EMPH by 7.3%. The mel-spectrogram modifications of MEL-EMPH degrades the quality and prosody compared to DD-EMPH, which integrates the duration modification within the neural TTS architecture. With DD-EMPH, the acoustic model is able to adapt the prosody based on seen examples in the training set to match the requested phoneme duration. Note that for this voice, reducing the factor $\alpha_{DD}$ to 1.25 of DD-EMPH to match MEL-EMPH (see Figure 6) still shows 3% performance improvement for DD-EMPH over MEL-EMPH.

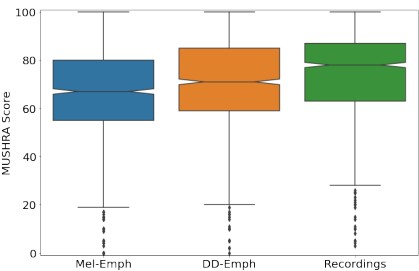

Figure 5: DD-EMPH *with* $\alpha_{DD} = 1.5$ *improves naturalness over* MEL-EMPH *with* $p < 0.0001$ *according to a Friedman test. Average MUSHRA score are 65.8 for* MEL-EMPH, *70.6 for* DD-EMPH *and 74 for Recordings.*

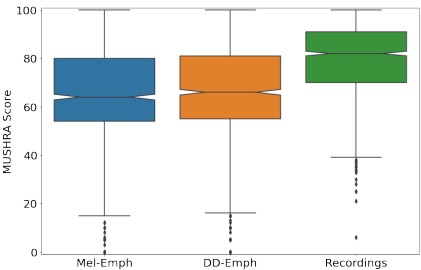

Figure 6: DD-EMPH *with* $\alpha_{DD} = 1.25$ *improves naturalness over* MEL-EMPH *with* $p < 0.0001$ *according to a Friedman test. Average MUSHRA score are 64.2 for* MEL-EMPH, *66.1 for* DD-EMPH *and 79.3 for Recordings.*

When comparing to the MUSHRA results presented for DD-EMPH (see Figure 5) and FLAG-EMPH (see Figure 3), FLAG-EMPH shows an extra absolute improvement of $5.4\% (= 12.7\% - 7.3\%)$ on top of DD-EMPH. The preference test shown in Table 2 confirms that if emphasis data are available, modelling emphasis with an encoder significantly improves performance.

Table 2: FLAG-EMPH *is strongly preferred over* DD-EMPH ($\alpha_{DD} = 1.5$).

| | Voice | Δ **Pref.** | $p$**-value** |
|---|---|---|---|
| en-US | female-0 exp. | $-25.6\%$ | $< 0.001$ |

We observe on Table 3 that word emphasis identifiability for this voice is improved with DD-EMPH ($\alpha_{DD} = 1.5$) over MEL-EMPH by $40\%$ on the en-US voice. This is due to two effects: (1) the duration with DD-EMPH is further increase by a factor 0.25, (2) the acoustic model is adapting the prosody to match the increased length by making that word stand out more.

We also tried for MEL-EMPH to further increase the word emphasis naturalness and identifiability by increasing the factor to $\alpha_{mel} = 1.5$ from 1.25 and loudness to $V_{mel} = 1.3$ from 1.15. A preference test between the two showed strong preference for the initial set of parameters ($\alpha_{mel} = 1.25$ and $V = 1.15$). Identifiability was increased at the cost of audio quality and naturalness.

Table 3: *Identifiability test: Emphasised words are more identifiable with* DD-EMPH *($\alpha_{DD} = 1.5$) than with* MEL-EMPH.

| | Voice | MEL-EMPH | DD-EMPH |
|---|---|---|---|
| en-US | female-0 exp. | $43\%$ | $60\%$ |

### 3.4. Reproducibility study

We run a reproducibility study on 6 additional voices divided across 4 locales with results shown on Table 5. We observe that DD-EMPH is strongly preferred for the voices trained on more expressive data. As pointed out above, our model is able to associate duration changes with other acoustic measures of emphasis when the training data is very expressive, providing the model a sufficient number of cases of emphatic speech. When the training data for the target voice are neutral, performance is degrading. When listening to the samples, we observe that emphasised word with DD-EMPH on models trained on neutral data makes the word sound long and somewhat unnatural. In other words, the model is making the duration change but is not making any additional association with pitch contour changes and does not add any additional silence as in the cases discussed in Section 3

### 3.5. Acoustic analysis of a case of DD-EMPH

This section aims to explain why an approach based solely on duration modification, specifically making all phonemes in a word longer by a certain factor, would produce perfectly emphasized words in most cases. We focus on the analysis of a single case, as representative of many other similar ones. We used the Praat software [32] to compare three acoustic dimensions (duration, pitch, energy) for the word *traditionally* when emphasized by our model and when produced without emphasis (see Figure 7), as part of the sentence *and it's traditionally one of the experiences we naturally try to avoid*. We observed that the duration is as expected longer for the emphasised version. Pitch and intensity are however quite similar in both cases and, if anything, maximum pitch and higher intensity are in fact slightly higher for the non-emphasized version of the word, as detailed in table 4. This suggests that a rise in pitch or energy are not absolutely necessary for the perception of emphasis (see [20] for a similar conclusion on Swedish with respect to pitch). Notice though that the difference between minimum and maximum pitch is slightly higher for the emphasized word, which relates to the particular pitch contour obtained for the word.

The model however appears to have in fact implicitly learned two aspects of emphatic speech that it was not explicitly trained on:

1. The role of silence preceding the syllable carrying primary stress (see [2]): a silence preceding this syllable is clearly visible when the word is emphasized (figure 7b), but not when the word is not (figure 7a).

2. The contour of f0 shows a clear rise in figure 7b, but is essentially flat in figure 7a. Moreover, the pitch gets to a L point much faster in the case of emphasis. We take this contour to instantiate well-known H*+L contour, associated with narrow focus in classical studies like [33], [34] and much subsequent work.

We conclude that the model has implicitly associated duration lengthening with emphasis in this case (and many similar ones). We present data below suggesting that our approach is particu-

larly successful on voices build from highly expressive recordings, while it does not work as well on voices built from 'neutral' recordings. Evidently, in order for the model to be able to implicitly associate emphasis and phoneme lengthening, there needs to be a sufficient number of such cases in the training data. This is borne out in the case of highly expressive data, but not in the case of neutral data.

An additional point confirming this hypothesis is that the model appears to work sub-optimally in the case of unstressed monosyllabic words (prepositions, determiners, etc.). These are unlikely candidates for emphasis and essentially absent in emphasized form in training data. The model is thus incapable of associating duration lengthening and emphasis in such cases. It is worth noting that monosyllabic words that are more likely to occur as emphasized in training data work as expected under our approach (for example the word *not*).

Our study shows that even if not all properties usually associated with focus are present in the signal, emphasis is still perceived. We suggest that increased phoneme duration, a rise-fall in pitch contour and a short silence before the emphasized portion of speech suffice to convincingly trigger the perception of emphasis.

Table 4: *Numerical values for energy and pitch for the word* traditionally *when emphasized and non-emphasized.*

| Emphasis | No emphasis | Measure |
|---|---|---|
| 153.4 Hz | 169.2 Hz | mean pitch |
| 118.4 Hz | 127.1 Hz | minimum pitch |
| 229.9 Hz | 236.0 Hz | maximum pitch |
| 69.6 dB | 70.5 dB | mean-energy intensity |
| 75.5 dB | 75.8 dB | maximum intensity |

Table 5: *Preference tests comparisons between* DD-EMPH *with* $\alpha_{DD} = 1.5$ *and* MEL-EMPH *emphasis.*

|  | Voice | $\Delta$ Pref. | $p$-value |
|---|---|---|---|
| en-US | male-0 conv. | 22.6% | < 0.001 |
| en-US | female-1 neutral | 0.8% | 0.700 |
| en-US | female-1 exp. | 4.0% | < 0.001 |
| es-US | female-2 neutral | −6.3% | 0.002 |
| es-US | female-3 neutral | 1.2% | 0.500 |
| es-US | female-3 expressive | 9% | < 0.001 |
| de-DE | female-4 neutral | −24.4% | < 0.001 |
| it-IT | female-5 conv. | −11.4% | < 0.001 |

To compensate for this effect, we decided to reduce for these voices the $\alpha_{DD}$ of DD-EMPH to 1.25 to be comparable to MEL-EMPH. As shown on Table 6, it significantly improves the preference of DD-EMPH over MEL-EMPH for these voices. We believe that this is due to speech quality degradation brought by the speech signal processing technique.

### 3.6. Does DD-EMPH emphasis improve over no emphasis?

So far, we have made the assumption that modifying the speech produced with DD-EMPH does not degrade speech quality and has some positive effect on the produced speech. In Table 7, we show that DD-EMPH is preferred by the listeners to no-emphasis for 4 voices.

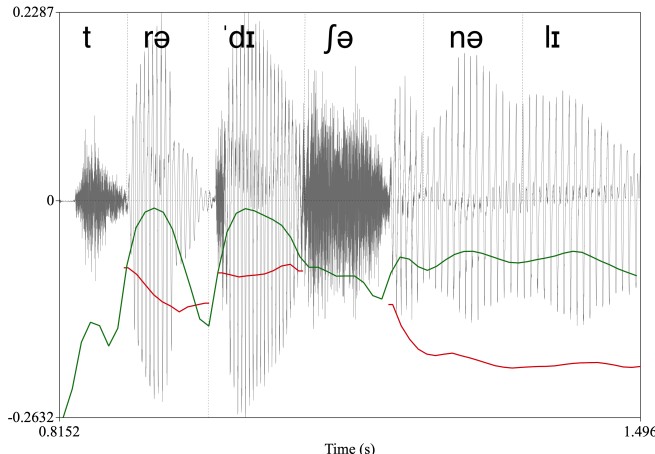

(a) Word **traditionally** when non-emphasized

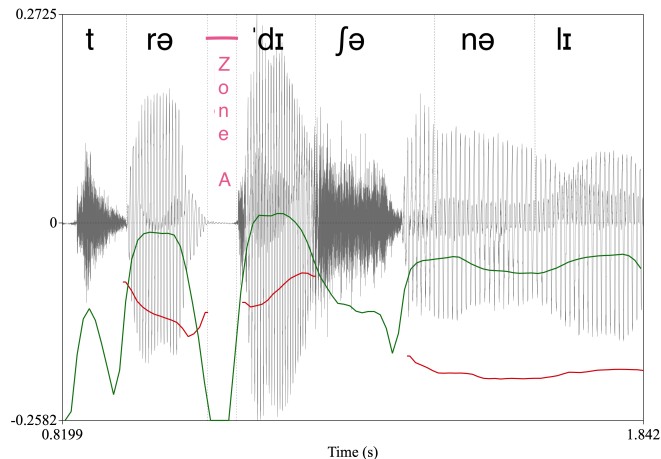

(b) Word **traditionally** when emphasized

Figure 7: *Pitch (red), intensity (green), waveform (grey) for the word* **traditionally** *in two versions of the sentence "and it's traditionally one of the experiences we naturally try to avoid." In the zone A, we are observing a longer phoneme duration resulting in a longer closure and a hard stop.*

Table 6: *Preference tests comparisons between* DD-EMPH *with* $\alpha_{DD} = 1.25$ *and* MEL-EMPH *emphasis*

|  | Voice | $\Delta$ Pref. | $p$-value |
|---|---|---|---|
| en-US | female-1 exp. | 5.2% | < 0.001 |
| es-US | female-2 neutral | 5.3% | < 0.001 |
| es-US | female-3 neutral | 1.9% | 0.060 |
| de-DE | female-4 neutral | 6.8% | < 0.001 |
| it-IT | female-5 conv. | 1.8% | 0.070 |

## 4. Conclusions

We have shown that it is possible to build a controllable word emphasis system without requiring recordings, annotation or re-training, and without degrading quality. We have leveraged the decoupling of duration and acoustic models in a non-attentive deep learning TTS model to bring emphasis by dilating dura-

Table 7: *Preference tests between DD-emph and no emphasis.*

| Voice | | $\alpha_{DD}$ | Δ **Pref.** | $p$-**value** |
|---|---|---|---|---|
| en-US | female-0 exp. | 1.5 | 11.4% | < 0.001 |
| en-US | male-0 conv. | 1.5 | 5.6% | 0.006 |
| en-US | female-1 exp. | 1.25 | 7.0% | < 0.001 |
| es-US | female-3 exp. | 1.5 | 9.0% | < 0.001 |

tion (DD-Emph) of target emphasised words. Our DD-Emph approach improves quality by 7.3% and identifiability by 40% over the mel-spectrogram modification baseline (Mel-Emph). It is scalable in multiple voices, locales and styles. We believe this approach is applicable for non attentive TTS system where the acoustic model is driven by a duration model.

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
