# OpenReview forum: "Controllable Emphasis with zero data for text-to-speech"
_Interspeech.org/2023/Workshop/SSW — SSW12_

### Official Review · Reviewer_JnVm · 2023-05-22
**Nice idea but sloppy evaluation**

**Rating:** 6
**Confidence:** 5

**Review:**

The authors compare methods to emphasize words in neural TTS that use a phoneme duration predictor. They compare a system trained with data with explicit emphasis (where the speaker is instructed to emphasize certain words) with a system trained on data with no explicit labelling of emphasis, for which the durations of the word with emphasis are just multiplied by a certain factor. They demonstrate that the covariations between prosodic dimensions learned by the acoustic decoder significantly improve perceived naturalness over spectrogram modification techniques.

* Key Strength of the paper :
Control of emphasis that does not require explicit recordings nor labelling

* Main Weakness of the paper :
Evaluation methodology is weak, since subjects are asked to rate utterances with emphasis. Acoustic analysis of only one case is not sufficient to demonstrate the "naturalness" of the control policy!

* Novelty/Originality, taking into account the relevance of the work for the SSW audience :
This research is data-frugal

* Technical Correctness, is the work technically and/or scientifically solid? Are sufficient details provided to allow any experiments to be reproduced or equivalent experiments run?
Some details are clearly missing:
1. How test words with emphasis are chosen? some words with strong semantic weight are often emphasized by default (negations, superlatives, etc). What happens with these words?
2. The low anchor should be TTS without any emphasis control. The evaluation is biased by attention: the target word is explicitly given as instruction
3. How to interpret reference tests : do they test the acoustic degradation or the adequacy of the control policy?
4. Provide more stats on acoustic signatures: a case example is not a proof of concept.
Comments:
1. Please refer to word-internal "pauses" as hiatus!!!
2. For FLAG-EMPH, did you compare emphatic markers surrounding words under emphasis as an alternative to flag concatenation? Seems that one flag does not compete so well with the large dimension of phoneme embeddings...

* Quality of References, is it a good mix of older and newer papers? Do the authors show a good grasp of the current state of the literature? Do they also cite other papers apart from their own work?
Refs are adequate

* Clarity of Presentation, the English does not need to be flawless, but the text should be understandable
The text is clear enough

---

### Official Review · Reviewer_GLBW · 2023-06-04
**The authors present a simple-yet-effective approach, denoted as Duration Dilatation Emphasis, to emphasize specific words in text-to-speech (TTS) synthesis systems with non-attentive architectures.  As the results are very promising, providing some extra insights of the critical elements of the proposal could make the paper more robust and the proposal more comprehensive.**

**Rating:** 8
**Confidence:** 4

**Review:**

The authors present a simple-yet-effective approach, denoted as Duration Dilatation Emphasis, to emphasize specific words in text-to-speech (TTS) synthesis systems with non-attentive architectures. The key idea is focused on increasing the duration of the target emphasized word before creating the corresponding Mel-spectrograms instead of modifying them afterwards. Moreover, the model also derives the previous pause and the F0 rising contour on the emphasized syllable typically found when modelling this kind of syllables from recording speech, thus closing the gap to those models derived from real speech.

As the results are very promising, providing some extra insights of the critical elements of the proposal could make the paper more robust and the proposal more comprehensive.

Specific Comments
Abstract: “The method proved to be effective in all four languages tested” -> Please, include the main conclusions related to the generalization capabilities of your proposal on other languages different than English (Spanish, Italian and German), as done for female en-US voice.

Datasets: Where those datasets come from? Please, provide the corresponding sources.

Regarding the DD-EMPH model:

1. The duration model consists of a stack of 3 convolution layers with 512 channels, kernel size of 5 and a dropout of 30%, a Bi-LSTM layer and a linear dense layer.”  -> Please, briefly describe the experiments conducted to determine the key features of your duration model.
2. Since α_{DD} ∈[1.0, 1.5] -> As expressed, this is a continuous interval of values that allows the TTS system to increase the duration of specific phoneme by 50% at most, right? Hence, it would be interesting to explain or depict the distribution or most common α_{DD} values found, mainly to be compared with the ones the authors empirically derived from the Mel-Emph baseline approach.

3.It would be interesting to see the distribution of the duration model by phoneme or by group of phonemes (e.g., vowels, fricatives, plosives, etc.) to show the reader that even though, a priori, it makes no sense to uniformly modify the duration of plosives, for instance, your results are robust thanks to the contents of the training data.

As an idea, you could fuse MUSHRA figures depicting boxplots into a single figure to have room to include another figure showing these, let’s say, critical phonemes (denoted in your paper as stops and affricates).

Experiments: Did you consider any control point and/or metric to control the consistency of the 24 listeners who conducted the perceptual tests?

Minor comments
* References: “also known as phoneme-to-frame alignment [Refs],” -> Please include the corresponding reference
* Writing:
  “The acoustic model aims to predicts the mel-spectrogram …” -> aims to predict
  “Most TTS system generation are divided into” -> “Most TTS generation systems are…”?
  "acoustic model uniformy" -> uniformly
  "The voice talents is requested to bring narrow and focus emphasis on the emphasized word." -> The voice talent is..
   “as in the cases discussed in 3” -> in 3?
* Acronyms: Bi-LSTM definition, DD-emph in Table 7 caption.
* Recommendation: using only one decimal point to express the data in Table 4 (mainly for dBs)
*Bibliography:  Please, review references [28] and [31] as they contain some typos, and try to minimize/substitute those referred to arxiv, mainly those that have been published elsewhere e.g. [22] can be found at Interspeech2020: http://www.interspeech2020.org/index.php?m=content&c=index&a=show&catid=312&id=726, whose approach is denoted as DurIAN and not as DURIAN

---

### Decision · Program_Chairs · 2023-06-14

**Decision:**

Accept

**Comment:**

SSW2003 received 45 papers. The acceptance rate is 82%. We are pleased to inform you that your paper has been accepted by the SSW2023 Program Committee. Please read the reviews carefully and submit your camera-ready paper by June 28th. Most reviewers performed a detailed review. Please answer to their questions and consider their comments. Note that camera-ready papers are credited with one extra page to allow authors to consider reviewers’ suggestions. So max 7 pages in total including figures & refs.
The deadline for submitting the revised version (with full non-anonymized authors and refs!) is 28th June.